# A Unified Approach for Risk Assessment of Road Crash Barriers Using Bayesian Statistics

**Pavel Vrtal, Karel Kocián, Jakub Nováček, Zděněk Svatý and Tomáš Kohout \*,**

CTU in Prague, Faculty of Transportation Sciences, Department of Forensic Experts in Transportation, Konviktská 20, 110 00 Prague, Czech Republic

**\*** Correspondence: kohout@fd.cvut.cz

**Abstract:** The aim of this article is to improve road safety. Specifically, it deals with the development of a mathematical model that will more accurately define the severity of a defect in a road restraint system. Currently, that evaluation is based only on the subjective perception of individual safety auditors. The mathematical model was developed based on the principle of Bayesian statistics. The determination of the specific risk was made by comparing the results of the model for two datasets. In the first case, the model was based on accident data correlated with recorded defects in road restraint systems. In the second case, the dataset represented accident events with crash barriers where no defect was identified. Based on the comparison, a total of 64 risk combinations were identified. The mathematical model confirmed 26 combinations (41% of all selected combinations of the defect levels of the crash barriers). Although not even half of the identified combinations were confirmed, more than 90% of all correlated records are found in these exposures of confirmed combinations. The verification was able to clearly define the risk of safety defects and thus brings potential accuracy to subsequent decision-making related to the repair of road restraint systems.

**Keywords:** road safety; road accidents; road restraint systems; crash barriers; road safety inspection; Bayesian statistics; discrete modelling

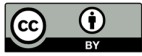

## 1. Introduction

The issue of traffic accidents is a long-standing problem that requires constant attention. The Czech Republic, as well as other EU countries, is required to adopt a plan to reduce the consequences of traffic accidents, known as the White Paper [1]. This document serves as a foundation for several national strategic documents, including the National Road Safety Strategy in the Czech Republic [2,3]. One of the main goals of current policy is to promote the "Vision Zero" idea [4], which represents a traffic system without deaths or serious injuries. While achieving this goal in the coming years may be merely utopian, it is still believed to be possible or that it will be possible to come close to doing so in the future [5].

The implementation of this policy requires comprehensive developments in the field of road safety. In order to describe a road as safe, it is necessary that the road fulfills two basic conditions—that its safety features are self-explanatory and flexible [6,7]. One of the tools to achieve this goal is Road Safety Inspections. This tool is integrated by legislation into practice in many countries around the world [8–13]. It is essential to improving road safety.

The principle of road safety is to identify the risk factors of the selected road section. Subsequently, measures are proposed to eliminate them. The inspection team can rate the identified risk factors with three levels of risk severity: low, medium, and high [8]. The rating helps the inspection client to prioritise when deciding whether to address risk factors, and if so, which ones to address and in what order of priority. The inspection team

determines the severity of the risk based on their qualifications and experience. The circumstances involved in the occurrence of accidents are complex, and estimating the level of safety risks identified is a challenging task [14,15].

However, the description of this risk rating scale is exceptionally general. According to the experience of road safety auditors and road managers, the severity of the same types of defects differs. The rating is particularly problematic in the area of restraints, as that is perceived differently by individual auditors who assess them subjectively. Specifically, this pertains to road restraint systems, which include, for example, crash barriers, crash cushions, and other safety features, as well as safety equipment of the road. The primary function of these features is to absorb energy when cars crash. The precondition for fulfilling this function is their absolute functionality, which can only be achieved if the road restraint systems are properly designed and implemented [16–19].

The primary objective of the manuscript is to explicitly standardize and subsequently validate the riskiness of the identified traffic safety deficits in the area of interceptor systems. The benefit of implementing a clearly defined methodology will allow road safety auditors to accurately rate the riskiness of restraint systems. The topic of this article is focused on this category because assessment of the riskiness of these systems from the point of view of professional practice is often a subject of discussion and a matter of opinion. In addition, the article can provide road managers with a tool to help them focus on assessing the riskiest defects and prioritising the reconstruction of selected objects. Thus, they will be able to make more economical use of the financial and human resources at their disposal and avoid investing more time on less serious problems that do not significantly affect the health consequences of a traffic accident [6,20]. The risks are referenced to the crew of an average passenger car.

## 2. Research on Current Scientific Knowledge

The topic of road safety, specifically road safety inspections, as well as other approaches to assessing the level of safety of the transport space, has been addressed by a number of domestic and foreign research studies. Some possible examples include a study from Alabama, USA, that assessed the influence of risk factors in relation to the severity of injuries caused by crashing into a crash barrier (i.e., vehicle detention, vehicle redirection, and barrier breach). A total of 1685 crash barrier accidents that occurred on three major international highways over a seven-year period (2010–2016) were analysed. Aspects of the barrier (e.g., central reservation width, barrier length, barrier offset, and lateral location) were assessed. Two types of longitudinal barriers were analysed: high tension cable barriers installed in the central reservation and fixed post barriers installed in the central reservation or soft shoulders. Separate mixed logit models (MXL) were used to analyse crash injuries, assessing the severity of crashes related to barrier impacts and estimating injury consequences. The MXL models were able to identify factors that contributed to crash severity and consequences of crashes with crash barriers. Recommendations were made that crash barriers longer than 0.2 miles (322 m) should be implemented to reduce the probability of barrier breaches and reduce the number of crashes with these consequences [21].

Another example is a study explaining human biomechanics in relation to guardrail intrusion into a vehicle's crew cab. At the same time, the study evaluates the biomechanical effectiveness of hybrid tensile compression guardrails for better passenger protection. Nine fatal crashes with guardrail intrusion into the vehicle were analysed. Four crash tests between cars and crash barriers were performed using a hybrid guardrail that integrated a commonly used W-beam with a new tension-based end section design. The test involved the impact of a platform colliding with small vehicles (sedan, pick-up) at highway speed. The impact orientation was varied to simulate frontal and curve impacts with speeds ranging from 90 to 111 km/h. Studies showed that fatal injuries were caused by guardrail impalement regardless of vehicle speed and size. Passengers who were not in the trajectory of the guardrail in the same vehicle sustained minor injuries despite experiencing

similar energy levels. In these cases, the impact intensity was survivable. The average pre-impact speed, change in speed, and acceleration of the vehicle were 117 km/h, 20 km/h, and 97 m.s$^{-2}$, respectively. The hybrid guardrail system deflected the vehicle without any intrusion into the passenger compartment. The average peak accelerations in the crash tests were below the injury thresholds. Research shows that the hybrid guardrail system not only eliminated intrusion into the passenger compartment for survival of the crash, but it also deflected the vehicle off the collision course [22].

Vehicle collision research with bridge crash barriers was presented in a study that focused on a new type of prefabricated crash barrier. This restraint system was assessed using LS-DYNA, a nonlinear imaging dynamic analysis program, and numerically analysed models of vehicle collisions with this type of crash barrier were developed. Subsequently, a time curve of the energy distribution during the vehicle collision was generated. Based on the developed model, the collision process of the vehicle with the installed anti-collision crash barrier was analysed. The result shows that the assembled anti-collision crash barrier proposed in this article can better change the trajectory of the moving vehicle and can prevent the vehicle from falling off the bridge. The collision results show that the assembled anti-collision crash barrier for bridges proposed in this article can reduce vehicle damage and effectively protect the driver. Based on the repeated tests, it can be seen that the new type of prefabricated anti-collision crash barrier has good protective properties even under different working conditions [23].

A study similar to the above-mentioned article addresses the use of mechanical analysis and finite element simulation technology to evaluate the crash barrier and to optimise its construction. A mechanical model of the vehicle and the crash barrier was developed to calculate the force applied when the vehicle impacts the crash barriers. The impact force model was then used to calculate the impact force of a ten-ton truck. The wall thickness and column spacing could be optimised by the results found. The simulation results show that the maximum lateral dynamic deformation of the barrier was 1818 mm, which is consistent with the actual vehicle test results (1600 mm). When the wall thickness of the barrier was increased by 2 mm, the maximum lateral dynamic deformation was 1419 mm; when the spacing of the reinforced columns was reduced to 15 m, the deformation was 1364 mm; after optimization, the deformation was reduced by more than 20%, and the crashworthiness of the barrier was obviously improved [24].

Studies related to the analysis of restraint systems are also carried out on a traffic driving simulator, investigating how drivers adapt their driving trajectory when going through curves where diverse types and heights of barriers are located. The results of the research confirm that the height of the crash barrier has a significant effect on the lateral separation of the vehicle from the device and the impact on the vehicle's control. At minimum crash barrier height, drivers stay closer to the shoulder, while higher crash barriers result in drivers increasing their lateral distance. In the study, the speed of travel around these barriers was simultaneously evaluated, and speed was influenced by both barrier geometry and human factors. An interesting finding was that people adapted differently to the limitation of available sight distance caused by the crash barriers. Men drive faster and behave more aggressively than women [25].

Furthermore, the above study is related, for example, to the identification of the exact type of crash barriers using laser scanning of the road. After the point cloud acquisition, the segmentation method of binary coding was used to voxelize and recognize the crash barriers in the highway scene by using the cluster cutting method. Use of this method makes it possible to distinguish the types of barriers based on their characteristic cuts. The experimental results show that it can effectively identify the types of crash barriers in the point cloud with high accuracy [26]. The next study aims to modify the point cloud obtained from laser scanning and significantly reduce the size of this point cloud to improve further processing and object recognition. Then, segmentation and classification of the crash barriers are proposed as processes dependent on geometric parameters. The results show good discriminative ability in terms of classification compared to other modern

methods. Better results were achieved for steel crash barriers than for concrete crash barriers. The method was tested on a set of point clouds obtained by a mobile laser scanner from conventional roads and highways [27].

The results of other studies related to crash barriers are mainly professional works that result in manuals, guidelines and other recommendations for road managers, which encourage the construction or reconstruction of the road network with regard to increasing the safety of its users (e.g., SAFESIDE [28]). Furthermore, there are projects (e.g., SAVeRS [29]) that discuss the appropriateness of using diverse types of restraint systems (crash barrier with beginning element or terminal x crash cushion).

Last but not least, there is also research dealing with the selection of appropriate traffic safety improvement measures (i.e., eliminating identified traffic safety defects) based on economic evaluation of effectiveness (e.g., IRDES [30]). Finally, it is necessary to mention a project like road safety inspection, namely, the International Road Safety Assessment Programme iRAP [31], which, however, does not allow the determination and illustrative verification of the riskiness of a specific road safety defect on roads. The output of this programme is the overall riskiness of a 100 m section and not the specific severity of the defect.

A partially similar project has been implemented in the past for the fixed obstacle category [32]. It focused on the problem of determining the severity of fixed obstacles as a function of their exposure, also using the consequences of road accidents to verify its hypothesis. Differences in this work relate to the fact that hitting a fixed obstacle is completely undesirable. In contrast, road restraint systems are primarily designed to be struck by road users in the event of an accident (e.g., an impending run off a high embankment of road) by a vehicle.

The above-mentioned publications presenting the issues related to the analysis of road restraint systems mostly deal with topics not related to this research. The research found no studies addressing a standardised assessment of the safety risks with restraint systems.

## 3. The Principle of the Mathematical Model

The mathematical model is based on Bayesian statistics [33–35]. It works on the principle of discrete model classification and works with different combinations of solved data sets. Each combination contains a number of traffic accidents (traffic accidents related to the correlated accident counts of each type of restraint system defect), which are represented by the resulting society-wide loss. These data are discrete values that can only take on single values from a predefined interval.

The validity of the use of Bayesian modelling is demonstrated by the use of this approach in thematic research related to traffic safety. Bayesian modelling within a scientific approach to traffic safety has been used, for example, to assess conflict extremes in real time or to model traffic accidents on an urban road network [36,37].

The Bayes formula can be expressed as follows:

$$f(A, C) = \frac{f(A|B, C)f(B|C)}{f(A|C)},$$ (1)

The main significance of the Bayes formula is the conversion of the a priori probability density f(B|C) to the a posteriori probability density f(B|A,C). The a priori probability density describes only the random variable B as a function of the random variable C. In contrast, the a posteriori probability density also uses information from the random variable A, using the probability density f(A|B,C).

In general, the most common regression model is a linear regression model of the random component (noise) based on the following general equation:

$$y_t = \psi'_t \theta + e_t,$$ (2)

When augmented with a dynamic regression model and regression parameter vector, it is written as follows:

$$y_t = b_0 u_t + a_1 y_{t-1} + b_1 u_{t-1} + .. + a_n y_{t-n} + b_n u_{t-n} + k + e_t, \tag{3}$$

The regression discrete model simultaneously contains all input variables with a finite number of values. By observing the modelled variable in time, described by the dynamic model, it can be observed at time t its prediction at time t + k, or its value $y_{(t+k)}$ can be predicted. Based on the conditional probability density, a description of the prediction can be expressed:

$$f(y_{t+k}|y(t)), \tag{4}$$

The mathematical model is implemented using the Scilab program [38,39]. It calculates the probability for each combination of the crash barrier based on the society-wide damage from traffic accidents caused by three different types of defects (short turned down end of the crash barrier, incorrect beginning/end of the crash barrier, insufficient minimum distance behind the crash barrier). The output of the model's algorithm gives the probabilities for each combination.

The results are analyzed and compared for each defect individually. For each type of defect, the average society-wide loss for each combination is provided. Tables comparing the results from the mathematical model are then presented, showing abbreviated versions with only the combination number that determines the exposure of the crash barrier and the results. Blank cells in the tables represent combinations for which no accident events are included in the model.

## 4. The Method

The input data for the solution of this issue was the database of registered traffic safety defects managed by the Czech Roads and Motorways Directorate [40] and at the same time the database of traffic accidents registered by the Czech Police. For a comprehensive monitoring period of five years (2015 to 2019), a total of 10,198 accidents were recorded on the road network in the Czech Republic [41], for which the type of collision was determined by the police from the category of collision into a restraint system. Of this set of crashes, almost 60% (5987 traffic accidents) were identified on the TEN-T road network [42].

Since it is obvious that the crash barrier is designed to be hit by cars in the event of an accident, the risk level of the crash barriers cannot be calculated based only on recorded accidents at the location of identified crash barrier defects. The determination of the risk level is therefore carried out based on the comparison of the results of a mathematical model designed to compare the two following cases: For the first set of data, the type of mathematical model was filtering input data based on the mutual correlation of traffic accidents and recorded defects in the area of crash barriers. In the second case, the model is based on crash barrier accidents where no defect was located in the road safety inspections. Therefore, no other defects that could negatively affect safety in the area could be recorded in the vicinity of these accident events. A total of 5528 deficiencies in crash barriers were identified in six regions of the Czech Republic (approximately half of the area of the Czech Republic). The mutual relationship between the data set of traffic accidents and identified traffic safety defects of crash barriers was located.

This resulted in correlated traffic accidents that are influenced by a traffic-safety deficit from the category of restraint systems. The most common type of deficit is "Short beginning element of the crash barrier", which was correlated with crash events in 101 cases (Figure 1). The second most common (80 records) is "Inadequately realized beginning element of the crash barrier" (for example, the crash barrier does not have the beginning element sufficiently secured in the ground or there is no beginning element) (Figure 2).

"Failure to maintain minimum clear zone behind the crash barrier" was identified in 76 correlated records (Figure 3).

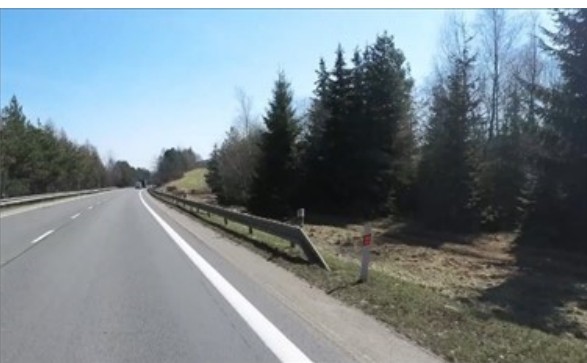

**Figure 1.** Short turned down end of the crash barrier.

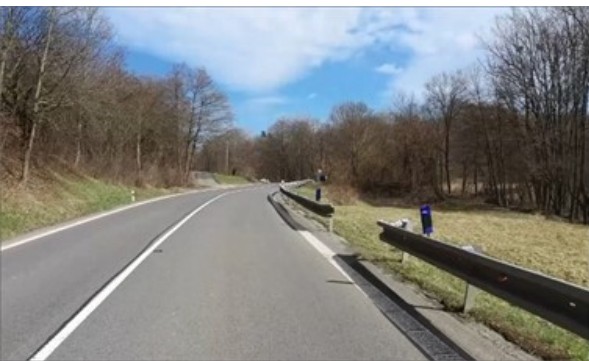

**Figure 2.** Incorrect beginning/end of the crash barrier.

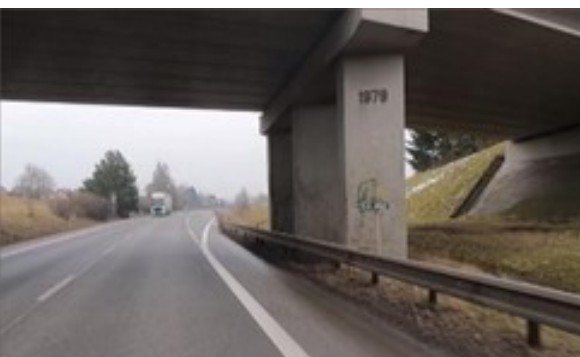

**Figure 3.** Insufficient minimum distance behind the crash barrier.

These three categories represent almost 80% of the entire dataset. The remaining restraint system defects are divided into a further eight categories, but their frequency is negligible and cannot be used for further mathematical evaluation due to the small number of correlated records.

In order to determine the theoretical risk for specific road restraint system defects, it is necessary to define the characteristic features of the adjacent road and the parameters of the crash barrier. The assignment of the parameters then generates the individual combinations that the correlated record can acquire. For these combinations, the mathematical model verifies the resulting riskiness of the recorded defect and is able to verify or disprove the hypothesis.

Traffic safety defects, for which parameters are defined, have various effects on traffic accidents. Therefore, the selected parameters are not identical for all defects, but may vary. (Table 1) The basic properties of the crash barrier and its immediate surroundings that are monitored are:

**Table 1.** Assigned parameters according to the type of defect.

| | Level of Restraint of the Crash Barrier | Directional Deflection of the Start of the Crash Barrier | Distance of the Crash Barrier from a Fixed Obstacle | Maximum Speed Limit | Spatial Road Alignment | Position of the Defect Relative to the Direction of Travel |
|---|---|---|---|---|---|---|
| Short turned down end of the crash barrier | | X | | X | X | X |
| Incorrect beginning/end of the crash barrier | | X | | X | X | X |
| Insufficient minimum distance behind the crash barrier | X | | X | X | X | |

Each of the deficits contains a different number of assigned parameters depending on their specific properties. For example, when assessing the short turned down end, it is not necessary to have information about the level of restraint of the crash barrier (ability to absorb the impact depending on the amount of energy released by the colliding vehicle) or the distance of the crash barrier from a fixed obstacle (position of fixed objects in the area of the crash barrier). Otherwise, it is necessary to know, for example, whether the start of the crash barrier is directionally deflected.

It can be seen from the table that the first two defects (short turned down end, incorrect beginning/end of the crash barrier) are defined by the same parameters. This is mainly due to the same characteristics of the defect, where in both cases the deficit is related to the incorrect beginning of the crash barrier. In contrast, the deficit "insufficient minimum distance behind the crash barrier" is usually recorded out of the beginning of the barrier; therefore, the parameters defining the design of the crash barrier at its beginning are not used. The remaining parameters (maximum permissible speed, spatial routing) are applied to all types of deficits, as both speed and routing are always important.

A more detailed classification of the individual parameters is presented in Table 2 below:

**Table 2.** Numerical indication of the value of selected parameters.

| Parameter Name | Parameter Name | ID |
|---|---|---|
| Level of restraint of the crash barrier [43] | N2 | 1 |
| | H1 | 2 |
| | H2 | 3 |
| Directional deflection of the start of the crash barrier | No direction deflection | 1 |
| | Partial direction deflection | 2 |
| Distance of the crash barrier from a fixed obstacle | (0; ½ working width) | 1 |
| | (½ working width; end of working width) | 2 |

| | Inappropriate road alignment | 1 |
|---|---|---|
| Spatial road alignment | Appropriate road alignment | 2 |
| Position of the defect relative to the | Right | 1 |
| direction of travel | Left | 2 |

The characteristic parameters for the barrier and its surroundings have been defined, and these determine its exact exposure. In addition, individual accidents must also be categorized based on the society-wide loss. The most appropriate option seems to be to split the accidents into three groups, based on the severity scale from the road safety inspections [8]. This scale can be used to rate potential road safety defects. The resulting grouping of the accidents according to society-wide traffic accident loss is based on the dataset of correlated records, since the society-wide accident loss in this dataset is only affected by the defects.

Figure 4 shows the distribution of society-wide traffic accident losses (sorted in ascending order), where a significant part of it describes an exponential curve. Approximately 25% of the data (exceedingly high loss accidents) then cease to fit this distribution and there is a more pronounced increase in loss. This is mainly due to the recorded grave consequences (severe injuries, more minor injuries) on the health of the people in these traffic accidents. At the same time, almost 40% of the accidents have a registered society-wide loss of up to 200,000 CZK. In the case of these accidents, a very gradual, initially almost linear, increasing trend can be observed.

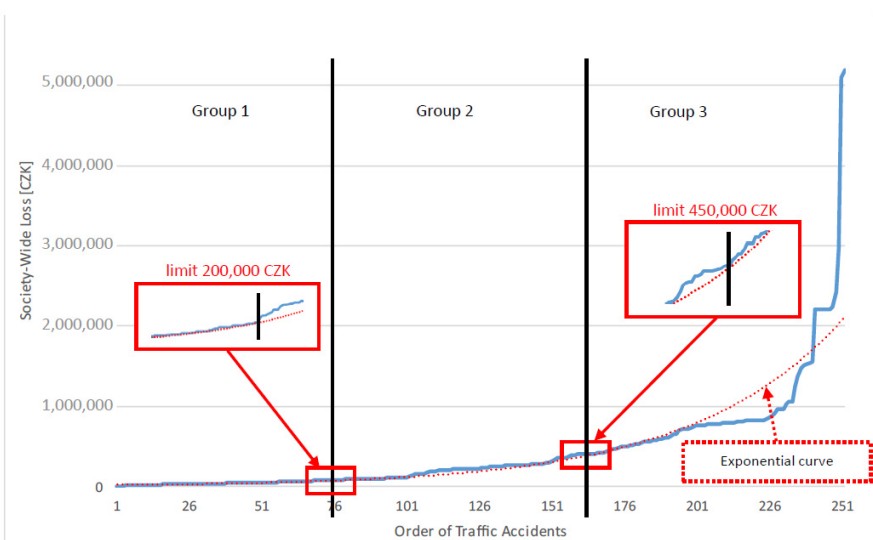

**Figure 4.** Distribution of Society-Wide Loss [CZK] for correlated traffic accident records.

On the resulting exponential curve, locations where there is a significant deviation from the exponential curve were located. This deviation means that there is a significant increase in the society-wide loss from traffic accidents. Such sharp increases in the resulting loss occur several times in the graph, but the two most significant increases (ignoring the last 25% of the data, which do not follow the exponential trend) were located at the thresholds of 200,000 CZK and 450,000 CZK (8100 EUR and 16,200 EUR). These values were chosen as the thresholds for the grouping of traffic accidents. Table 3 shows the resulting range of groups into which traffic accidents are divided according to the society-wide loss.

**Table 3.** Distribution of traffic accidents into selected groups depending on the society-wide loss.

| ID | Corresponding Risk | Range of Society-Wide Loss |
|----|--------------------|-----------------------------|
| 1 | Low Risk | ⟨0–200 000⟩ |
| 2 | Medium Risk | ⟨200 000–450 000⟩ |
| 3 | High Risk | ⟨450 000 and more⟩ |

The methodology for road safety inspections [8] provides the basic assumptions for determining the risk level of individual restraint systems defects. Together with these assumptions, it is possible to use the approximate risk level for each combination of defects based on the society-wide loss shown in the previous table. It is essential to note that the resulting assessment is currently based on subjectivity only, not on a uniform way of defining this severity.

Within the framework of the research, three hypotheses were defined, according to which the evaluation of individual combinations will be conducted:

The first hypothesis rests on the assumption that the stated risk level of each exposure of the traffic safety defect "short turned down end of the crash barrier" corresponds to the actual severity for the passenger car crew.

The second hypothesis aims to test whether the stated level of risk of each exposure of the traffic safety defect "incorrect beginning/end of the crash barrier" corresponds to the actual severity for the passenger car crew.

The third hypothesis verifies whether the stated level of risk of each combination of the traffic safety defect "insufficient minimum distance behind the crash barrier" correspond to the actual severity for the passenger car crew.

The verification of the hypotheses is carried out by comparing the results of the mathematical model in the form of a comparison of the outputs of the dataset of crash barriers accidents, which occurred at locations where no safety defects were recorded, and the dataset of correlated accidents with identified defects. The predicted severity of each defect with each combination of input variables can be seen in the following Tables 4–6. The predicted risk level is based on the methodology for road safety inspections [8] and at the same time on the knowledge base of the author's team resulting from long-term expert experience.

**Table 4.** Predicted risk level of individual exposures for defect—Short turned down end of the crash barrier.

| Combinations for Individual Speeds | Directional Deflection of the Start of the Crash Barrier | Position of the Defect Relative to the Direction of Travel | Spatial Road Alignment | Maximum Speed Limit | |
|---|---|---|---|---|---|
| | | | | 70 km/h | 90 km/h |
| 1., 2. | 1 | 1 | 1 | Medium Risk | High Risk |
| 3., 4. | 1 | 1 | 2 | Medium Risk | Medium Risk |
| 5., 6. | 1 | 2 | 1 | Low Risk | Medium Risk |
| 7., 8. | 1 | 2 | 2 | Low Risk | Low Risk |
| 9., 10. | 2 | 1 | 1 | Low Risk | Low Risk |
| 11., 12. | 2 | 1 | 2 | Low Risk | Low Risk |
| 13., 14. | 2 | 2 | 1 | Low Risk | Low Risk |
| 15., 16. | 2 | 2 | 2 | Low Risk | Low Risk |

**Table 5.** Predicted risk level of individual exposures for defect—Incorrect beginning/end of the crash barrier.

| Combinations for Individual Speeds | Directional Deflection of the Start of the Crash Barrier | Position of the Defect Relative to the Direction of Travel | Spatial Road Alignment | Maximum Speed Limit | | |
|---|---|---|---|---|---|---|
| | | | | 50 km/h | 70 km/h | 90 km/h |
| 1., 2., 3. | 1 | 1 | 1 | Medium Risk | High Risk | High Risk |
| 4., 5., 6. | 1 | 1 | 2 | Medium Risk | High Risk | High Risk |
| 7., 8., 9. | 1 | 2 | 1 | Medium Risk | Medium Risk | High Risk |
| 10., 11., 12. | 1 | 2 | 2 | Low Risk | Medium Risk | High Risk |
| 13., 14., 15. | 2 | 1 | 1 | Low Risk | Medium Risk | High Risk |
| 16., 17., 18. | 2 | 1 | 2 | Low Risk | Medium Risk | Medium Risk |
| 19., 20., 21. | 2 | 2 | 1 | Low Risk | Low Risk | Medium Risk |
| 22., 23., 24. | 2 | 2 | 2 | Low Risk | Low Risk | Medium Risk |

**Table 6.** Predicted risk level of individual exposures for defect—Insufficient minimum distance behind the crash barrier.

| Combinations for Individual Speeds | Level of Restraint of the Crash Barrier | Distance of the Crash Barrier from a Fixed Obstacle | Spatial Road Alignment | Maximum Speed Limit | |
|---|---|---|---|---|---|
| | | | | 70 km/h | 90 km/h |
| 1., 2. | 1 | 1 | 1 | High Risk | High Risk |
| 3., 4. | 1 | 1 | 2 | High Risk | High Risk |
| 5., 6. | 1 | 2 | 1 | Medium Risk | High Risk |
| 7., 8. | 1 | 2 | 2 | Medium Risk | High Risk |
| 9., 10. | 2 | 1 | 1 | Medium Risk | High Risk |
| 11., 12. | 2 | 1 | 2 | Medium Risk | High Risk |
| 13., 14. | 2 | 2 | 1 | Low Risk | Medium Risk |
| 15., 16. | 2 | 2 | 2 | Low Risk | Medium Risk |
| 17., 18. | 3 | 1 | 1 | Low Risk | Medium Risk |
| 19., 20. | 3 | 1 | 2 | Low Risk | Medium Risk |
| 21., 22. | 3 | 2 | 1 | Low Risk | Medium Risk |
| 23., 24. | 3 | 2 | 2 | Low Risk | Low Risk |

The above combinations of parameters for individual defects always contain a specific risk—no combination was rated as "No Risk". This assumption is primarily based on the high safety requirements for road restraint systems, where a faultless technical condition and methodologically correct construction are required for correct function.

**5. The Hypothesis Validation**

*5.1. The Hypothesis I.—Short Turned Down End of the Crash Barrier*

Hypothesis I. establishes the severity for combinations of "Short turned down end of the crash barrier" defects. Table 7 shows the increase in the society-wide loss that occurs in traffic accidents around the recorded defects of road restraint systems, where the table clearly shows a significant increase in values.

**Table 7.** Intercomparison of the average society-wide loss due to defect—Short turned down end of the crash barrier.

| Combinations | Uncorrelated Records | | Correlated Records | |
|:---:|:---:|:---:|:---:|:---:|
| | 70 km/h | 90 km/h | 70 km/h | 90 km/h |
| 1., 2. | 105,917 | 241,622 | 165,000 | 501,311 |
| 3., 4. | 93,438 | 223,372 | 95,250 | 475,967 |
| 5., 6. | 109,833 | 208,807 | 40,000 | 338,696 |
| 7., 8. | 78,350 | 237,340 | 33,000 | 287,487 |
| 9., 10. | | 216,683 | | |
| 11., 12. | | 84,091 | | 148,700 |
| 13., 14. | | 30,667 | | |
| 15., 16. | | | 10,000 | |

Table 8 already describes the results of the mathematical model for the dataset of the solved traffic safety defect.

**Table 8.** Mutual comparison of the probability of each combination for the defect—Short turned down end of the crash barrier. (L = Low Risk, M = Medium Risk, H = High Risk).

| Combinations | Uncorrelated Records | | | | | | Correlated Records | | | | | |
|:---:|:---:|:---:|:---:|:---:|:---:|:---:|:---:|:---:|:---:|:---:|:---:|:---:|
| | 70 km/h | | | 90 km/h | | | 70 km/h | | | 90 km/h | | |
| | L | M | H | L | M | H | L | M | H | L | M | H |
| 1., 2. | 83% | 17% | 0% | 68% | 10% | 22% | 100% | 0% | 0% | 22% | 22% | 56% |
| 3., 4. | 69% | 31% | 0% | 77% | 8% | 15% | 100% | 0% | 0% | 29% | 43% | 29% |
| 5., 6. | 100% | 0% | 0% | 80% | 0% | 20% | 100% | 0% | 0% | 41% | 44% | 15% |
| 7., 8. | 100% | 0% | 0% | 80% | 0% | 20% | 100% | 0% | 0% | 33% | 61% | 6% |
| 9., 10. | | | | 67% | 17% | 17% | | | | | | |
| 11., 12. | | | | 100% | 0% | 0% | | | | 67% | 33% | 0% |
| 13., 14. | | | | 100% | 0% | 0% | | | | | | |
| 15., 16. | | | | | | | 100% | 0% | 0% | | | |

According to the previous findings, it was possible to establish individual sub-conclusions:

- **The hypothesis was confirmed—There are sufficient records in the datasets for the combinations, and at the same time the stated assumption corresponds to the resulting probability.**

A high severity was established in the hypothesis for the 2nd combination, which was confirmed by comparing the results of the mathematical model. The probability of elevated risk for this combination increased by 34% to 56%. At the same time, the probability of medium risk increased, but only by 12% (to 22% overall). The low risk decreased by almost 50%.

Furthermore, the hypothesis for the 4th and 6th combination was confirmed. A medium risk was proposed in the hypothesis for these crash barrier exposures, which is consistent with the model results.

Finally, the 7th and 12th combinations were confirmed. In the case of the first combination, all recorded cases were rated only low risk. However, for the 12th combination, the medium risk has increased by 33% (previously it was zero), but the resulting probability for low risk is still double.

- **The hypothesis was confirmed—There are not enough records in the datasets for the combinations, but the resulting probability of a small number of data is consistent with the stated assumption.**

From the exposures for 70 km/h, the 5th combination was confirmed. It contains only one recorded case in the set of correlated records, but its probability shows the correctness of the determined risk based on expert knowledge.

- **The hypothesis has been modified. There are sufficient records for the combinations in the datasets, but the stated assumption does not correspond to the resulting probability:**

The 8th combination according to the results does not correspond to the proposed hypothesis, where a low risk was determined for this combination. Based on the results, there is a 61% higher probability for medium risk in the case of correlated records, at a resulting 61%. The probability for low risk then decreases by almost 50% to a resulting value of 33%. Based on the mathematical model, it is therefore necessary to adjust the hypothesis for this combination to a medium risk.

- **The hypothesis was not confirmed—There are not enough records in the datasets for the combinations and at the same time the assumption does not correspond to the resulting probability.**

The hypothesis in this case is neither confirmed nor refuted (due to the small number of records). More measured data is needed to verify it.

For combinations 1 and 3, only two cases are located in the correlated records. The resulting risk (determined on the basis of probability) for the dataset of correlated records is lower than the hypothesis. This bias in the results is due to the low, and therefore insufficient, number of records.

- **The hypothesis was not confirmed—There is no record for the combinations in the data files, and the stated assumption cannot be confirmed.**

The remaining combinations (9th–11th and 13th–16th) could not be verified due to lack of data in the datasets. The very fact that fewer traffic accidents are recorded for these combinations indicates that they are not very risky combinations.

Hypothesis I. was confirmed in eight cases out of 16 combinations. A total of 8 exposures of the crash barrier were not confirmed. Table 9 below shows the resulting confirmed combinations. Hypothesis I. is hereby verified and is valid for combinations 2, 4–8, 12 and 15.

**Table 9.** Hypothesis I.—Short turned down end of the crash barrier.

| Combinations | Directional Deflection of the Beginning of the Crash Barrier | Position of the Defect Relative to the Direction of Travel | Spatial Road Alignment | Maximum Speed Limit | |
|---|---|---|---|---|---|
| | | | | 70 km/h | 90 km/h |
| 1., 2. | 1 | 1 | 1 | Unconfirmed | High Risk |
| 3., 4. | 1 | 1 | 2 | Unconfirmed | Medium Risk |
| 5., 6. | 1 | 2 | 1 | Low Risk | Medium Risk |
| 7., 8. | 1 | 2 | 2 | Low Risk | Medium Risk |
| 9., 10. | 2 | 1 | 1 | Unconfirmed | Unconfirmed |
| 11., 12. | 2 | 1 | 2 | Unconfirmed | Low Risk |
| 13., 14. | 2 | 2 | 1 | Unconfirmed | Unconfirmed |
| 15., 16. | 2 | 2 | 2 | Low Risk | Unconfirmed |

### 5.2. The Hypothesis II.—Incorrect Beginning/End of the Crash Barrier

Hypothesis II establishes the severity for the defect combination "Incorrect beginning/end of the crash barrier". Table 10 shows the increase in society-wide loss that occurs in traffic accidents around recorded road restraint systems defects. The effect of road safety equipment defects on the amount of society-wide loss is demonstrated. In the right part of the table for the correlated records, a significant increase in the average values is clearly visible. Only in the case of the 15th parameter combination of the crash barrier

there is a decrease, which is primarily due to the limited amount of data in the correlated dataset for this exposure.

**Table 10.** Intercomparison of the average society-wide loss due to defect—Incorrect beginning/end of the crash barrier.

| Combinations | Uncorrelated Records | | | Correlated Records | | |
|---|---|---|---|---|---|---|
| | 50 km/h | 70 km/h | 90 km/h | 50 km/h | 70 km/h | 90 km/h |
| 1., 2., 3. | 41,714 | 105,917 | 241,622 | 215,000 | 505,333 | 805,210 |
| 4., 5., 6. | 51,300 | 93,438 | 223,372 | 60,000 | 446,750 | 586,900 |
| 7., 8., 9. | 24,000 | 109,833 | 208,807 | | | 517,667 |
| 10., 11., 12. | 55,000 | 78,350 | 237,340 | | 277,000 | 480,950 |
| 13., 14., 15. | 35,429 | | 216,683 | | | 57,333 |
| 16., 17., 18. | | | 84,091 | | | |
| 19., 20., 21. | | | 30,667 | | | 45,000 |
| 22., 23., 24. | | | | | | |

Table 11 describes the results of the mathematical model for both datasets of the solved traffic safety defect. The evaluation of the probability comparison is again divided into groups based on the result of the hypothesis verification.

**Table 11.** Mutual comparison of the probability of each combination for the defect—Incorrect beginning/end of the crash barrier. (L = Low Risk, M = Medium Risk, H = High Risk).

| Combinations | Uncorrelated Records | | | | | | | | | Correlated Records | | | | | | | | |
|---|---|---|---|---|---|---|---|---|---|---|---|---|---|---|---|---|---|---|
| | 50 km/h | | | 70 km/h | | | 90 km/h | | | 50 km/h | | | 70 km/h | | | 90 km/h | | |
| | L | M | H | L | M | H | L | M | H | L | M | H | L | M | H | L | M | H |
| 1., 2., 3. | 100% | 0% | 0% | 83% | 17% | 0% | 68% | 10% | 22% | 0% | 100% | 0% | 0% | 33% | 67% | 0% | 20% | 80% |
| 4., 5., 6. | 100% | 0% | 0% | 69% | 31% | 0% | 77% | 8% | 15% | 100% | 0% | 0% | 0% | 25% | 75% | 17% | 0% | 83% |
| 7., 8., 9. | 100% | 0% | 0% | 100% | 0% | 0% | 80% | 0% | 20% | | | | | | | 0% | 33% | 67% |
| 10., 11., 12. | 100% | 0% | 0% | 50% | 0% | 50% | 80% | 0% | 20% | | | | 0% | 100% | 0% | 17% | 33% | 50% |
| 13., 14., 15. | 100% | 0% | 0% | | | | 67% | 17% | 17% | | | | | | | 100% | 0% | 0% |
| 16., 17., 18. | | | | | | | 100% | 0% | 0% | | | | | | | | | |
| 19., 20., 21. | | | | | | | 100% | 0% | 0% | | | | | | | 100% | 0% | 0% |
| 22., 23., 24. | | | | | | | | | | | | | | | | | | |

- **The hypothesis was confirmed—There are sufficient records in the datasets for the combinations, and at the same time the stated assumption corresponds to the resulting probability.**

  High risk was confirmed in the 2nd hypothesis for the 3rd, 6th, 9th and 12th combinations for 90 km/h respectively for the 2nd and 5th combinations for 70 km/h.

  For the first group, the probability for high risk was increased by almost 60% on average, with a similar decrease in low risk. The probability for medium risk in these cases increased only slightly. The exception is the 12th combination, where there was a 33% in medium risk and 30% increase in high risk. Compared to this change, however, the overall probability of high risk remains significantly higher at 50%—the medium risk has a 33% probability.

  In the case of the second group (speed 70 km/h) for the 2nd combination the probability increased by 67%, and for the 5th combination the probability increased by 75%. In both cases, there was a zero probability of high risk in the dataset excluding the recorded accidents.

- **The hypothesis was confirmed—There are not enough records in the datasets for the combinations, but the resulting probability of a small number of data is consistent with the stated assumption.**

Despite the low number of recorded cases, the hypothesis for the 1st and 11th combinations was confirmed, with these exposures being suggested as medium risk. The probability of this risk increased by 100% when comparing the datasets, as all recorded cases of correlated accidents belonged to this category. The probability determined from the small number of data corresponds to the established risk defined by expert knowledge.

- The hypothesis was not confirmed—There are not enough records in the datasets for the combinations and at the same time the assumption does not correspond to the resulting probability.

According to the results, the 4th, 15th, and 21st combinations do not correspond to the proposed hypotheses, where a medium risk was determined for this combination, but based on the comparison of the results of the mathematical model, the risk is low. This is due to the small number of records for these three combinations, where only three cases are recorded for the 15th combination and only one case is recorded for the remaining combinations.

- **The hypothesis was not confirmed—There is no record for the combinations in the data files, and the stated assumption cannot be confirmed.**

The remaining combinations (7th, 8th, 10th, 13th, 14th, 16th–20th, 22nd–24th) could not be verified due to lack of data in the datasets. The very fact that fewer traffic accidents are recorded for these combinations suggests that they are not very risky combinations.

The hypothesis II. was confirmed in eight cases out of 24 combinations. A total of 16 exposures of the crash barrier were not confirmed. The following Table 12 shows the final confirmed combinations. Hypothesis II. is hereby verified and is valid for combinations 1–3, 5, 6, 9, 11 and 12.

**Table 12.** Hypothesis II.—Incorrect beginning/end of the crash barrier.

| Combinations | Directional Deflection of the Beginning of the Crash Barrier | Position of the Defect Relative to the Direction of Travel | Spatial Road Alignment | Maximum Speed Limit | | |
| --- | --- | --- | --- | --- | --- | --- |
| | | | | 50 km/h | 70 km/h | 90 km/h |
| 1., 2., 3. | 1 | 1 | 1 | Medium Risk | High Risk | High Risk |
| 4., 5., 6. | 1 | 1 | 2 | Unconfirmed | High Risk | High Risk |
| 7., 8., 9. | 1 | 2 | 1 | Unconfirmed | Unconfirmed | High Risk |
| 10., 11., 12. | 1 | 2 | 2 | Unconfirmed | Medium Risk | High Risk |
| 13., 14., 15. | 2 | 1 | 1 | Unconfirmed | Unconfirmed | Unconfirmed |
| 16., 17., 18. | 2 | 1 | 2 | Unconfirmed | Unconfirmed | Unconfirmed |
| 19., 20., 21. | 2 | 2 | 1 | Unconfirmed | Unconfirmed | Unconfirmed |
| 22., 23., 24. | 2 | 2 | 2 | Unconfirmed | Unconfirmed | Unconfirmed |

*5.3. The Hypothesis III.—Insufficient Minimum Distance behind the Crash Barrier*

The hypothesis III establishes the severity for combinations of the defect "Insufficient minimum distance behind the crash barrier". Table 13 shows the difference in how much the society-wide loss increases for traffic accidents around recorded road restraint system defects. The trend of increasing society-wide loss from the bottom left corner to the top right corner is also confirmed here based on a peer comparison. At the same time, the effect of the traffic safety defect of the steel crash barrier on the society-wide loss is evident. The correlated records in the left part of the table achieve significantly higher average losses.

**Table 13.** Intercomparison of the average society-wide loss due to defect—Insufficient minimum distance behind the crash barrier.

| Combinations | Uncorrelated Records | | Correlated Records | |
|---|---|---|---|---|
| | 70 km/h | 90 km/h | 70 km/h | 90 km/h |
| 1., 2. | 70,000 | 162,922 | 50,000 | 780,550 |
| 3., 4. | 63,556 | 247,659 | | 1,598,125 |
| 5., 6. | 70,000 | 162,922 | | 577,311 |
| 7., 8. | 63,556 | 247,659 | | 580,425 |
| 9., 10. | 49,029 | 212,312 | | 95,000 |
| 11., 12. | 46,083 | 185,696 | 200,000 | 661,000 |
| 13., 14. | 49,029 | 212,312 | 42,000 | |
| 15., 16. | 46,083 | 185,696 | 50,000 | 219,005 |
| 17., 18. | 28,333 | 217,064 | 28,000 | |
| 19., 20. | 48,036 | 136,722 | | 85,000 |
| 21., 22. | 28,333 | 217,064 | | |
| 23., 24. | 48,036 | 136,722 | | |

Table 14 already shows the results of the mathematical model for both datasets of the solved traffic safety defect.

**Table 14.** Mutual comparison of the probability of each combination for the defect—Insufficient minimum distance behind the crash barrier. (L = Low Risk, M = Medium Risk, H = High Risk).

| Combina-tions | Uncorrelated Records | | | | | | Correlated Records | | | | | |
|---|---|---|---|---|---|---|---|---|---|---|---|---|
| | 70 km/h | | | 90 km/h | | | 70 km/h | | | 90 km/h | | |
| | L | M | H | L | M | H | L | M | H | L | M | H |
| 1., 2. | 100% | 0% | 0% | 81% | 7% | 12% | 100% | 0% | 0% | 10% | 30% | 60% |
| 3., 4. | 100% | 0% | 0% | 84% | 1% | 15% | | | | 25% | 25% | 50% |
| 5., 6. | 100% | 0% | 0% | 81% | 7% | 12% | | | | 22% | 33% | 44% |
| 7., 8. | 100% | 0% | 0% | 84% | 1% | 15% | | | | 13% | 38% | 50% |
| 9., 10. | 100% | 0% | 0% | 91% | 0% | 9% | | | | 75% | 25% | 0% |
| 11., 12. | 100% | 0% | 0% | 71% | 12% | 18% | 0% | 100% | 0% | 0% | 25% | 75% |
| 13., 14. | 100% | 0% | 0% | 91% | 0% | 9% | 100% | 0% | 0% | | | |
| 15., 16. | 100% | 0% | 0% | 71% | 12% | 18% | 100% | 0% | 0% | 17% | 83% | 0% |
| 17., 18. | 100% | 0% | 0% | 65% | 18% | 18% | 100% | 0% | 0% | | | |
| 19., 20. | 100% | 0% | 0% | 84% | 11% | 5% | | | | 100% | 0% | 0% |
| 21., 22. | 100% | 0% | 0% | 65% | 18% | 18% | | | | | | |
| 23., 24. | 100% | 0% | 0% | 84% | 11% | 5% | | | | | | |

- **The hypothesis was confirmed—There are sufficient records in the datasets for the combinations, and at the same time the stated assumption corresponds to the resulting probability.**

Based on the comparison of the results of the mathematical model, the hypothesis for the 2nd, 4th, 6th, 8th and 12th combinations with the predicted high severity was confirmed. In these cases, there is an average increase in the probability of high severity of 30% to 60% and medium severity of 25% on average. For these combinations, the probability for high risk is always the highest, almost always exceeding 50%. In the case of the 12th combination, the probability is as high as 75%. The probability for medium risk is significantly lower and in none of the cases exceeds 40%.

A medium level of severity was confirmed for the 16th combination, with more than 80% probability. This increased by 72% compared to the dataset with no recorded safety

features defects. The probability for this combination of high risk was zero and 17% for low risk.

Furthermore, the hypothesis for the 13th and 17th combinations was proved. Low severity was confirmed for these combinations, with no other cases identified for any other severity level.

- **The hypothesis was confirmed—There are not enough records in the datasets for the combinations, but the resulting probability of a small number of data is consistent with the stated assumption.**

The 11th and 15th combinations correspond to the severity levels of the hypothesis, i.e., medium (11th combination) and low (15th combination) risk. However, only one case was recorded for each of these exposures.

- **The hypothesis was not confirmed—There are not enough records in the datasets for the combinations, and at the same time the assumption does not correspond to the resulting probability.**

The following three combinations do not correspond to the hypothesis based on a comparison of the mathematical model outputs. These are the 1st, 10th and 20th combinations. In the case of the 10th combination, a high risk has been proposed; however, the model output has the highest probability for a low risk, namely 75%. However, there is a slight increase in the medium risk (25%). This is again due to the small number of recorded data—there are only four records for this exposure.

A comparable situation applies to the 1st and 20th combination, where only one case is recorded for each exposure. These cases then do not belong to the expected severity; their risk level is significantly lower.

- **The hypothesis was not confirmed—There is no record for the combinations in the data files, the stated assumption cannot be confirmed.**

The remaining combinations (3rd, 5th, 7th, 9th, 14th, 18th, 19th, 21st–24th) could not be verified due to lack of data in the datasets. The very fact that fewer traffic accidents are recorded for these combinations suggests that they are not very risky combinations.

Hypothesis III was confirmed from 24 combinations in 10 cases. A total of 14 exposures of the crash barriers were unconfirmed. Table 15 below shows the resulting confirmed combinations. Hypothesis III is hereby verified and is valid for combinations 2, 4, 6, 8, 11–13, 15–18.

**Table 15.** Hypothesis III.—Insufficient minimum distance behind the crash barrier.

| Combinations | Level of Restraint of the Crash Barrier | Distance of the Crash Barrier from a Fixed Obstacle | Spatial Road Alignment | Maximum Speed Limit | |
|---|---|---|---|---|---|
| | | | | 70 km/h | 90 km/h |
| 1., 2. | 1 | 1 | 1 | Unconfirmed | High Risk |
| 3., 4. | 1 | 1 | 2 | Unconfirmed | High Risk |
| 5., 6. | 1 | 2 | 1 | Unconfirmed | High Risk |
| 7., 8. | 1 | 2 | 2 | Unconfirmed | High Risk |
| 9., 10. | 2 | 1 | 1 | Unconfirmed | Unconfirmed |
| 11., 12. | 2 | 1 | 2 | Medium Risk | High Risk |
| 13., 14. | 2 | 2 | 1 | Low Risk | Unconfirmed |
| 15., 16. | 2 | 2 | 2 | Low Risk | Medium Risk |
| 17., 18. | 3 | 1 | 1 | Low Risk | Unconfirmed |
| 19., 20. | 3 | 1 | 2 | Unconfirmed | Unconfirmed |
| 21., 22. | 3 | 2 | 1 | Unconfirmed | Unconfirmed |
| 23., 24. | 3 | 2 | 2 | Unconfirmed | Unconfirmed |



## 6. Discussion

The total number of records for both types of datasets significantly impacts the accuracy and proper interpretation of the results. In order to thoroughly test the hypothesis, it is necessary to have a sufficient amount of data for each combination of defects, both for the dataset of traffic accidents outside of the recorded defects and for the correlated records. It has been concluded that the minimum number of cases that will provide acceptable predictive value is five records [44], but it is desirable to identify as many records as possible for each crash exposure in order to achieve higher accuracy of the output model.

Another uncertainty is the definition of the individual parameters characterising the properties of the crash barrier and its surroundings. The definition of these parameters is affected by a certain uncertainty. A variable that affects the resulting riskiness may have been omitted. To ensure that the authors minimized the influence of uncertainty on parameter selection, a broad discussion was held within the team of experts on the issue. At the same time, the influence of the selected parameters was repeatedly tested as part of the calibration of the mathematical model, comparing the hypothesis with the outputs of the mathematical model for each type of defect. As part of the verification, the validity of the defined risks for each combination in each hypothesis was verified.

## 7. Conclusions

The main idea of this study was to develop a methodological procedure that would refine the outputs of the road safety inspections carried out, specifically the defects related to road restraint systems. Following this procedure will ensure that identical results are always obtained from different investigators, in terms of assessing the level of severity of identified defects relating to crash barriers, crash cushions, and other road safety features. There will be no situations where different teams of road safety auditors assign different risks to the same defect. This will always allow the road manager to focus on eliminating the actual highest risk defects and thus reduce road accidents more effectively. At the same time, economic and human resources will be used in a meaningful way.

This fact means that the risk level cannot be calculated simply on the basis of recorded accidents at the location of the identified road restraint systems defects. The severity determination was therefore made by comparing the results of the mathematical model for the two datasets. In the first case, the model was based on accident data correlated with recorded road restraint defects (328 records). In the second case, it was based on crashes with crash barriers where no defect was located within the road safety inspection (627 records).

The research was focused on the determination and verification of the risk for the following three types of defects: short turned down end of the crash barrier, incorrect beginning/end of the crash barrier, insufficient minimum distance behind the crash barrier.

The severity of traffic accidents was assessed based on the societal loss from traffic accidents in three groups, according to the amount of societal loss caused. In order to determine the risk for a particular failure of a road restraint system, it was necessary to define the parameters of its surroundings and the characteristics of a given crash barrier that influence the resulting risk. In total, six types of parameters were identified that adequately define the exposure of a safety device: the level of restraint of the crash barrier, the directional deflection of the start of the crash barrier, the distance of the crash barrier from a fixed obstacle, the maximum permissible speed, the spatial road alignment, and the position of the defect relative to the direction of travel.

A specific risk level is proposed for each exposure characterised by the risk parameters. This procedure has been applied for all traffic safety defects analyzed. Thus, the hypotheses I.–III. were defined, where based on the comparison of the results of the mathematical model or the determined probabilities for each combination of defects, the subsequent verification of the hypotheses was performed. A total of 64 combinations were determined within the three road interceptor defects addressed. Of these, 26 combinations,

or approximately 41% of all exposures, were verified by the mathematical model. For the remaining 38 cases, a sufficiently informative data sample was not available. Although these are not even half of the confirmed combinations, more than 90% of all correlated records are found in these exposures with model-confirmed severity. Thus, these are the most frequently occurring exposures of traffic safety deficits on the road network. The findings clearly demonstrated the validity and appropriateness of the methodological approach chosen in the verification of the hypotheses.

Future research will continue to verify the remaining exposures that were not demonstrated during this analysis. Efforts will continue to develop a comprehensive and validated methodology for determining severity levels of traffic safety defects that can be used by the professional community and that will serve as a supportive tool for reducing accidents and loss of life.

**Author Contributions:** Conceptualization, J.N., P.V. and T.K.; methodology, Z.S.; software, Z.S., K.K; validation, J.N., P.V. and T.K.; formal analysis, T.K.; investigation, P.V. and K.K.; resources, P.V. and Z.S.; data curation, J.N.; writing—original draft preparation, J.N. and T.K.; writing—review and editing, P.V. and T.K.; visualization, J.N.; supervision, Z.S.; project administration, K.K. All authors have read and agreed to the published version of the manuscript.

**Funding:** This work was supported by the Grant Agency of the Czech Technical University in Prague, grant No. SGS20/199/OHK2/3T/16, SGS21/138/OHK2/2T/16 and SGS21/137/OHK2/2T/16.

**Institutional Review Board Statement:** Not applicable.

**Informed Consent Statement:** Not applicable.

**Data Availability Statement:** Not applicable.

**Conflicts of Interest:** The authors declare no conflict of interest.

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
