# Peer review of "A Unified Approach for Risk Assessment of Road Crash Barriers Using Bayesian Statistics"

_applsci, doi:10.3390/app13031270_

Round 1
Reviewer 1 Report
This study proposes an approach to assess the risk of crash barrier based on Bayesian framework. The reviewer has several concerns that need to be addressed before any publication.
1. Lines 22-25, these sentences are not clear.
2. What are the contributions of this study?
3. The reason why this study only focuses on road crash barriers is not presented.
4. section “2. Research” is the literature review?
5. What are the models based on the defect-related accidents and the crash barrier related accidents? They should be mentioned when first mentioned.
6. Several sentences are very difficult to understand. For example, “A correlation between the dataset of traffic accidents and identified road safety restraint systems defects was located”. This sentence is hard for the reviewer to understand.
7. what do you mean “assigned parameters according to the type of defect”
8. The used data and methodology are not presented clearly.
9. How to obtain the risk level of individual exposures of three defects (tables 4-6)?
10. The basic Bayesian framework can be removed.
11. The Bayesian approach is a model parameter estimation method, and it is not a model. With respect to the use of Bayesian approach, the authors can refer to these inferences such as: A multivariate method for evaluating safety from conflict extremes in real time; Bayesian spatial joint modeling of traffic crashes on an urban road network.
12. A dynamic linear model was developed? The reviewer is very confusing.
13. Why validate these three hypotheses? Why do not use the developed models for identifying the risky factors? The structure of the paper is confusing.
14. The writing needs more work, such as English and article structure.
15. Some professional terms/sentences are strongly suggested to add, otherwise it is hard to understand.
Reviewer 2 Report
The research divide the gap of road guardrail into three different types, allocate parameters for combination (parameters include the level of crash barrier, the starting deflection direction of crash barrier, the distance between crash barrier and the object to be impacted, the maximum speed limit, the linear space of the road, and the opposite direction of the gap and vehicle operation), and use the mathematical model based on Bayesian statistics to assess the risk level (divided into three levels), Compare the degree of risk increase of the three types of guardrail gap under different parameter combinations compared with the situation with guardrail, so as to find the three types of combination with the highest risk of guardrail gap, so as to give priority to the improvement of guardrail gap under this combination.
1. The 16 lines in the abstract mentioned the innovation of the risk assessment model in the article, but no specific description was found.
2. The research part is more like the accumulation of relevant research cases. The cited studies are not well connected, which seems quite abrupt. For example, in the cases of 107 rows and 120 rows, readers can only read the importance of guardrails for road safety. What kind of guardrails can be obtained from the model with stronger crash resistance? What is the relationship between the mathematical models listed and the mathematical models used in this paper?
3. Why does this paper choose the model based on Bayesian statistics? Lines 178-181 summarized the shortcomings of existing cases too simply, and did not explain the necessity of the article to provide risk assessment criteria.
4. I feel that the method and the principle of the mathematical model may read more smoothly if they are swapped.
5. How does the study divide the accident records related to the guardrail gap into such three specific types? I'm a little confused. Is there such a detailed record, or is there a field survey based on the data location?
Round 2
Reviewer 1 Report
The reviewer comments have been addressed.
Author Response
Thank you very much for the time that you invested in our manuscript. We found your comments very valuable and helpful and we are glad that we adequately addressed your comments.
Reviewer 2 Report
The author has made some modifications in the paper and marked them in highlighted mode, but it is not easy for reviewers to view them. It is suggested that a reply letter be provided to clarify the modification process and reply to review comments one by one.
